# Comparison of Mechanical Properties of Chairside CAD/CAM Restorations Fabricated Using a Standardization Method

**DOI:** 10.3390/ma14113115

**Published:** 2021-06-06

**Authors:** Myung-Sik Hong, Yu-Sung Choi, Hae-Hyoung Lee, Jung-Hwan Lee, Junyong Ahn

**Affiliations:** 1Department of Prosthodontics, College of Dentistry, Dankook University, Cheonan 31116, Korea; 0123carrotcake@gmail.com; 2Institute of Tissue Regeneration Engineering (ITREN), Dankook University, Cheonan 31116, Korea; haelee@dankook.ac.kr (H.-H.L.); ducious@dankook.ac.kr (J.-H.L.); ajy3809@naver.com (J.A.); 3Department of Nanobiomedical Science and BK21 PLUS NBM Global Research Center for Regenerative Medicine, Dankook University, Cheonan 31116, Korea; 4Department of Biomaterials Science, College of Dentistry, Dankook University, Cheonan 31116, Korea; 5UCL Eastman-Korea Dental Medicine Innovation Centre, Dankook University, Cheonan 31116, Korea; 6Cell & Matter Institute, Dankook University, Cheonan 31116, Korea

**Keywords:** chairside CAD/CAM restorations, mechanical property, fractography, fracture resistance, standardization

## Abstract

The aim of this in vitro study was to investigate the fracture resistance, fracture failure pattern, and fractography of four types of chairside computer-aided design/computer-aided manufacturing (CAD/CAM) restoration materials in teeth and titanium abutments fabricated using a standardization method. An artificial mandibular left first premolar prepared for all-ceramic crown restoration was scanned. Forty extracted mandibular molars and cylindrical titanium specimens were milled into a standardized shape. A total of eighty CAD/CAM restoration blocks were milled into a crown and twenty pieces of each lithium disilicate (LS), polymer-infiltrated-ceramic-network (PICN), resin nano ceramic (RNC), and zirconia-reinforced lithium silicate (ZLS) materials were used. Crowns were bonded to abutments, and all specimens underwent thermal cycling treatment for 10,000 cycles. Fracture resistance was measured using a universal testing machine and fracture failure patterns were analyzed using optical microscopy and scanning electron microscopy. Statistical differences were analyzed using appropriate ANOVA, Tukey HSD post hoc tests, and independent sample *t*-tests (α = 0.05). The results indicated that, in both teeth abutments and titanium abutments, the fracture resistances showed significantly the highest values in LS and the second highest in ZLS (*p* < 0.05). The fracture resistances based on teeth abutments and titanium abutments were significantly different in all the CAD/CAM restoration materials (*p* < 0.05). There are statistically significant correlations between the types of materials and the types of abutments (*p* < 0.05). Each of the different materials showed different fracture failure patterns, and there was no noticeable difference in fractographic analysis. Lithium disilicates and zirconia-reinforced lithium silicates exhibited statistically high fracture resistance, indicating their suitability as restoration materials for natural teeth or implant abutments. There were no distinct differences in the fracture pattern based on the restoration and abutment materials showed that the fracture initiated at the groove where the ball indenter was toughed and propagated toward the axial wall.

## 1. Introduction

With the rapid development of computer-aided design/computer-aided manufacturing (CAD/CAM) technology, dental CAD/CAM systems are actively used in the dental field today [1,2,3]. In particular, scanners and milling devices have improved significantly, and it is now possible to design and fabricate sophisticated dental restorations using CAD/CAM software [3,4]. This technology facilitates the design of specimens for in vitro studies that evaluate the function of dental materials. In addition, such technology is also used for standardizing and manufacturing in a specific form [5].

New dental prosthetic materials using dental CAD/CAM systems are also continuously being developed [1,6,7,8]. Using the chairside CAD/CAM material, it is possible to easily manufacture and install a prosthesis in a clinic, reducing the number of patient visits. A wide variety of aesthetic materials have been implemented for dental prosthesis using CAD/CAM systems, such as resin, ceramic, and zirconia [1,6,7,8]. Recently, hybrid CAD/CAM materials such as polymer-infiltrated-ceramic-network (PICN), resin nano ceramic (RNC), and zirconia-reinforced lithium silicate (ZLS) have been introduced [9,10,11,12,13,14]. PICN is manufactured in a three-dimensional double lattice structure by infiltrating a polymerized polymer between pre-sintered inorganic ceramic supports and is composed of 86 wt% inorganic ceramic and 14 wt% polymers [10,11]. RNC is a chemically bonded, nano-sized inorganic filler with a resin substrate. This is made of a resin ceramic polymer with a filler ratio of up to 80 wt% [12,13]. ZLS is a selectively soluble, zirconia-reinforced silicate that contains 10% dissolved zirconia in a silica-based glass matrix [14].

The mechanical properties of materials used in the fabrication of these chairside CAD/CAM restorations have been studied using various methods. To understand the mechanical properties of the material itself, most studies have been conducted by processing blocks in the form of cuboids or cylinders [9,14,15,16,17]. Artificial materials can be produced uniformly in the same shape, but human teeth cannot be unified as they all look different. To collect systematic information, it is necessary to standardize teeth as a specimen; however, standardizing teeth into the same shape requires complex processes, and studies that have attempted such procedures are insufficient. CAD/CAM materials can also be used for implant restorations. As implant treatment is actively performed, studies on CAD/CAM restorations in implant abutments are also actively progressing [18,19,20]. However, research comparing it with natural teeth is insufficient. In addition, studies on the fracture resistance of the upper structure due to the different elasticity modulus of the abutments are also scarce [21,22].

When analyzing the failure of a material, especially in ceramic materials, the analysis of the fracture pattern is of great significance. Therefore, we are interested in the analysis of fracture patterns and the fractography of these chairside CAD/CAM materials. In general, there is a difference that adhesion is used for teeth and cementation is used for titanium abutments [23,24]. However, few studies have compared this difference. Since fractography can only be observed after the material is fractured, there are few studies using in vivo restorations. As a preliminary step toward in vivo research, we attempted to analyze the fracture pattern under standardized conditions in vitro.

The aim of this in vitro study was to investigate the fracture resistances, fracture failure pattern, and fractography of four types of chairside CAD/CAM restoration materials—LS, ZLS, PICN, and RNC—in teeth abutments (TOs) and titanium abutments (TIs) fabricated using a standardization method.

The null hypotheses proposed in this study are as follows: first, there is no difference in the mechanical properties—fracture resistance, fracture failure pattern, and fractography—between four types of chairside CAD/CAM restorations cemented to TOs and TIs. Second, in each sample of the same CAD/CAM material, there is no difference in the mechanical properties between TO and TI. Third, there is no correlation in the mechanical properties between types of CAD/CAM materials and the types of abutments.

Overall, the main highlights of this study are to understand the correlations between the chairside CAD/CAM materials and abutments, and to investigate the clinical applicability of the materials as prosthetic restorations for teeth and implant abutments.

## 2. Material and Methods

A total of 40 extracted mandibular premolars with healthy crowns without caries or restorations were stored in a 5% NaOCl solution for less than 90 days (IRB No. DKUDH IRB 2020-07-004) [25]. The top of a VITA CAD-Temp multiColor (VITA Zahnfabrik, Bad Säckingen, Germany) block was uniformly cut—8.0 mm horizontally, 10.0 mm long, and 20.0 mm deep—and the root of the tooth was appropriately cut to the required depth. The lower part was fixed with Cyanoacrylate Adhesives (ALTECO Inc., Osaka, Japan), and the upper part was fixed with acrylic resin Miky Blue (NISSIN Dental Products Inc., Kyoto, Japan) to produce a block for a TO (Figure 1A,B).

A mandibular left first premolar resin tooth (NISSIN Dental Products Inc., Kyoto, Japan) was prepared, with an axial surface of 1.0 mm and an occlusal surface of 1.5 mm, in the form of an ideal abutment for all-ceramic crown restoration and scanned using the T500 scanner (Medit; Seoul, Korea). The form was duplicated as a standard tessellation language (STL) file (Figure 1C). After entering the STL file using CEREC inLab CAM (Dentsply Sirona, York, PA, USA) software, the teeth block was placed on a CEREC inLab MC XL (Dentsply Sirona, York, PA, USA) and milled for approximately 15 min using a Step Bur 20 (Dentsply Sirona, York, PA, USA) and Cylinder Pointed Bur 20 (Dentsply Sirona, York, PA, USA) (Figure 1D,E). The prepared abutment was stored in distilled water (Daihan Sterile Saline; Daihan Pharm, Seoul, Korea). To compare the mechanical properties of TOs and TIs, TIs were fabricated by milling 40 titanium blocks into the same form, achieved by CNC milling of premilled cylinder grade 5 titanium alloy (Ti-6Al-4V) with ARUM 5X-200 (Arum Europe GmbH, Frankfurt, Germany)—number of axes, 5; accuracy, 5 µm; spindle power DC, 3.0 KW; spindle speed, 2000–60,000 rpm; automatic tool changer (ATC) number of tools, 15 [26].

Four types of blocks were prepared as material for chairside CAD/CAM restorations: IPS e.max CAD (Ivoclar Vivadent, Schaan, Lichtenstein) as LS, VITA Enamic (VITA Zahnfabrik, Bad Säckingen, Germany) as PICN, Cerasmart (GC, Tokyo, Japan) as RNC, and Celtra Duo (Dentsply Sirona, York, PA, USA) as ZLS. The ingredients of the materials are shown in Table 1.

An artificial resin tooth of the mandibular left first premolar (NISSIN Dental Products Inc., Kyoto, Japan) was scanned using a model scanner T500 (Medit; Seoul, Korea) and duplicated as an STL file (Figure 1F). Using the CAD program Exocad v2.3-6990/64 (Exocad GmbH, Darmstadt, Germany), a crown shape was designed by superimposing the unprepared artificial tooth onto the prepared artificial tooth STL file, with the adhesive space set to 20 μm. After standardizing the crown shape using the CEREC inLab CAM software, four types of blocks were placed on the CEREC inLab MC XL and milled for approximately eight minutes, each using the Step Bur 12 and Cylinder Pointed Bur 12 (Dentsply Sirona, York, PA, USA). Post-treatment was performed on each crown according to the manufacturer’s instructions.

A total of 80 crowns were classified by material and the cementation process was carried out. Surface cleansing of crowns and abutments was performed. Then, the surface of the crowns and TIs were sandblasted with 50 μm alumina powder at an air pressure of 0.1–0.4 MPa (14–58 PSI), and they were cleaned using an ultrasonic device for 2 min, then dried with a stream of air. Surface pretreatment of crowns was performed according to the manufacturer’s instructions using Panavia F 2.0 (Kuraray, Tokyo, Japan). Clearfil Ceramic Primer (Kuraray, Tokyo, Japan) was applied to the internal surface of the restorations. Panavia F 2.0 ED Primer II (Kuraray, Tokyo, Japan) was applied to the surfaces of TOs. Additionally, mixed Panavia F 2.0 Paste was applied and the restorations were cemented with a pressure of 50 N using a Dynamometer (NK-200, HANDPI, Wenzhou, China) [27]. Excess paste was removed and finished (Figure 1G). The specimens were dried at room temperature for 24 h. The specimens were then classified into eight groups (N = 10 for each group).

Thermocycling treatment was performed 10,000 times (30 s at 5 °C; 30 s at 55 °C; rest period of 10 s) using a thermocycling machine (Tokyo Giken Co., Tokyo, Japan). This has been reported to be clinically equivalent to the condition of approximately 1 year in the oral environment [28]. After the treatment, all specimens were stored in distilled water at a room temperature of 20–25 °C.

The mechanical properties were evaluated through a fracture test. A jig that fixes the abutment in a certain position was fabricated so that a stainless-steel ball of diameter 3.0 mm could be evenly placed at the center of the occlusal surfaces of the crowns. The maximum fracture resistance (N) was measured by applying a load at a crosshead speed of 0.5 mm/min using an Instron 5966 (Instron Corporation, Canton, OH, USA) universal material tester [29,30].

The failure patterns were analyzed. Fracture surfaces were visually inspected, and the fracture patterns were classified into adhesive failures, cohesive failures, and mixed failures [29,30].

Fracture patterns and their causes were observed and analyzed. A representative specimen was selected, and the fracture surface was observed at 25× magnification with an optical microscope (S39B, MICroscopes INC., St. Louis, MO, USA). Fracture patterns were analyzed by photographing the fracture surface with a camera connected to the microscope (Nikon C-DSD230, Nikon, Tokyo, Japan) [18,31,32,33].

The surfaces to be analyzed by an optical microscopy were coated with gold using an E1010 ion sputter (Hitachi, Tokyo, Japan). Using a field emission scanning electron microscope (FE-SEM) (Sigma 300, ZEISS, Oberkochen, Germany), cracks were photographed and observed at a magnification that satisfactorily revealed their shapes from the lowest 21× magnification to the highest 336× magnification. By analyzing the images, detailed fracture patterns were observed and the causes of the fractures were identified [18,31,32,33].

## 3. Statistical Analysis

The maximum fracture resistance was statistically analyzed using SPSS v25.0 (IBM SPSS Inc., Armonk, NY, USA). The normality was tested using the Shapiro–Wilk test, and Levene’s equal variance test was also performed. To compare the fracture resistance of each group, the evaluation was performed using one-way ANOVA, and an independent sample t-test was used to compare the average value of the fracture resistance of the restoration based on the abutment. Two-way ANOVA was used to analyze the correlation between the restoration and the abutment, and the Tukey HSD post hoc test was performed as a post-mortem test—it was evaluated with a significance level of 95% (α = 0.05).

## 4. Results

### 4.1. Analyses of Fracture Resistance

The average values and standard deviations (SDs) of fracture resistance in the TO and TI groups of the four types of restorations are as follows (Table 2).

According to the one-way ANOVA analysis, in the TO group, LS (1137.33 ± 139.30 N) had the highest fracture resistance value, and the second was ZLS (976.47 ± 107.37 N) (*p* < 0.05). There was no statistically significant difference between PICN (789.73 ± 98.90 N) and RNC (707.39 ± 100.74 N) (*p* > 0.05; Figure 2A). In the TI group, LS (1346.60 ± 103.53 N) showed the highest value for fracture resistance, followed by ZLS (1211.32 ± 93.70 N), PICN (670.24 ± 40.80 N), with RNC (334.39 ± 36.30 N) showing the lowest value (*p* < 0.05; Figure 2B). According to the independent samples t-test, the fracture resistances based on TO and TI groups were significantly different in all the CAD/CAM restoration materials (*p* < 0.05). In the LS and ZLS groups, the fracture resistance values of TIs were significantly higher than the values of TOs (*p* < 0.05). Conversely, in the PICN and RNC groups, the fracture resistance values of TOs were significantly higher than the values of TIs (*p* < 0.05; Figure 3). Two-way ANOVA analysis revealed significant correlations between the four types of chairside CAD/CAM restorations and types of abutments. (*p* < 0.05; Table 3).

### 4.2. Fracture Failure Mode

The failure modes at the fracture surfaces were visually observed and classified into adhesive failures, cohesive failures, and mixed failures, as shown in the following graphs (Figure 4). Representative specimens are shown in Figure 5. Seven adhesive and three mixed failures in the LS_TO group, six adhesive and four mixed failures in the PICN_TO group, three adhesive and seven mixed failures in the RNC_TO group, and five adhesive and mixed failures in the ZLS_TO group were observed. In the LS_TI group, mixed failures were found in all ten specimens. One adhesive and nine mixed failures in the PICN_TI group, eight adhesive and two mixed failures in the RNC_TI group, four adhesive failures, and six mixed failures in the ZLS_TI group were observed.

Fracture failure patterns were different in each group. For PICN and LS, adhesive failures occurred predominantly in the TO groups, whereas mixed failures were predominantly observed in the TI groups. Moreover, when adhesive failures were observed in PICN, ZLS, and LS, separations between the abutment and the cement were observed in most situations, whereas separation between the crown and the cement were only observed in RNC.

### 4.3. Fractography Analysis

Representative specimens were selected, and the fractures and propagation patterns were observed at the fracture surfaces of the restoration at 25× magnification using an optical microscope (Figure 6). There were no clear differences in the fracture patterns based on the restoration materials and types of abutments. In all specimens, it was confirmed that the fractures propagated into the restoration at the point at which the ball indenter was in contact with it. Fragments started from the area in which the indenter touched the restoration and extended to the crown margin. No chippings or catastrophic fractures were observed. Most of the specimens fractured into one large piece and smaller pieces—in some specimens, large pieces were broken into two or three smaller pieces. In the area in which the indenter was in contact, the material was crushed and several small fragments were formed.

By observing the specimens with magnification (as specified) using a FE-SEM, each fracture pattern could be viewed and the origin of the fracture was confirmed (Figure 7). In all specimens, the hackle—the shape of the fracture line extending out—could be observed, and the origin of the fracture could be inferred from the point at which the hackle gathered. Two fracture mirrors were found in ZLS_TO and one fracture mirror in RNC_TI, indicating a clear origin of fracture. A fracture mirror refers to a flat surface near the origin of fracture, reported to be the basis for the most discernible origin [32]. In other specimens, the fracture mirror could not be observed; thus, it could be inferred that the fracture started at the point at which the stainless-steel ball touched the specimen.

## 5. Discussion

In this study, TIs and TOs were standardized using CAD/CAM technology, and the mechanical properties of four different chairside CAD/CAM materials were compared: LS, ZLS, PICN, and RNC. There are statistically significant correlations between the types of materials and the types of abutments (*p* < 0.05). Each of the different CAD/CAM materials showed different fracture failure patterns, and there was no noticeable difference in fractographic analysis.

Each individual oral environment varies significantly—temperature, humidity, eating habits, lifestyles, and teeth shapes—and these can affect the mechanical properties and lifespan of the restoration materials. When conducting a comparative study of mechanical properties, considering the individual’s oral environment is clinically important; however, there are limitations to performing the study with actual patients. First, the shapes of the teeth and restorations are not standardized, making results difficult to compare. In previous research using extracted teeth, teeth were prepared with an ideal thickness and shape according to the anatomical shape of each tooth, so the shapes were not standardized [29,30]. Second, fracture properties are difficult to study, because the researcher would have to wait a long time until the restoration fractures through human masticatory strength. In the previous studies, most restorations were observed after fracture and the cause was deduced for each case [17,34]. Consequently, in most studies that attempt to compare the mechanical properties of materials by fabricating a crown, the main model is used to reproduce a specimen by taking a silicon impression of one tooth and casting it [5,35,36]. Since this method has a complicated manufacturing process, errors may occur in the impression-taking and casting steps. In addition, the experiment cannot be reproduced in other studies unless the original main model is mechanically intact. Impression-taking using CAD/CAM technology can simplify traditional procedures, and scanned data can be used to reproduce specimens of the same shape any time and place. In most previous studies, the specimen itself was produced by milling blocks of artificial materials rather than natural teeth, so the oral environment could not be reproduced [5,36]. Therefore, in this study, we endeavored to manufacture a specimen using a standardized method by preparing natural teeth in a certain abutment shape with a tooth standardization technique using CAD/CAM technology. This method can accurately replicate different teeth in the same form and use natural properties—such as wettability, thickness, and the pulp pressure of dentin—that cannot be imitated in artificial materials [37,38]. In addition, it has been reported that the thermal cycling treatment performed on a specimen can obtain an effect resembling the body aging process as if it were a natural specimen [39].

As a result of analyzing the fracture resistance of various chairside CAD/CAM restorations adhering to the standardized TOs and TIs, the fracture resistances in the TO group were significantly different between all restorations except between PICN and RNC, and the fracture resistances in the TI group were significantly different between all restorations. Therefore, the first null hypothesis stating that there would be no difference in mechanical properties between four types of chairside CAD/CAM restorations was rejected. The second null hypothesis stating that there would be no difference in the mechanical properties of CAD/CAM materials based on the abutment types was rejected, as there were significant differences in the fracture resistances based on the type of abutment for the same material. Additionally, since there was a correlation between the restorations and the abutment materials, the third null hypothesis stating that there would be no correlation between the types of CAD/CAM materials and the types of abutments was also rejected.

In previous studies, it has been reported that the fracture resistances of restorations are affected by the differences in the elastic modulus of the material of the upper restoration and the abutment [40,41]. Furthermore, a highly rigid abutment can protect the restoration from fracture—for example, a ceramic crown cemented to a cast gold core showed a significantly lower fracture index than a crown cemented to a tooth [22,42]. If the results of these studies are applied, TIs should have higher fracture resistance than TOs for all upper restoration materials. PICN and RNC, however, exhibited the opposite results in this study. This may be because the differences in the elastic modulus of the resin-containing restoration materials, PICN and RNC, were smaller for the TOs than the TIs. According to previous studies, the elastic modulus was 95 GPa for LS, 70 GPa for ZLS, 30 GPa for PICN [43], 5–42 GPa for natural tooth dentin [44], and 100–110 GPa for titanium alloy [45].

A previous study has shown that resin CAD/CAM restorations have a 3–4 times higher fracture fatigue strength than ceramic restorations, so CAD/CAM restorations are more endurable [46]. Accordingly, since the fatigue strength of a resin-containing hybrid restoration is higher than that of ceramic restoration, higher tensile stress is generated on the tooth—of which the elastic modulus is significantly lower than that of titanium—thereby increasing the fracture resistance of the resin-containing restoration. It is thought that the smaller the difference in the elastic modulus between TO and the upper restoration, the less the stress concentration at the interface between the restorations, resulting in the stress being distributed to a wider part of the crown [40]. Consequently, highly rigid ceramic materials such as ZLS and LS can resist higher stresses when supported by a rigid substructure, such as a TI. Conversely, in materials with a low elastic modulus similar to teeth, such as PICN and RNC, the stress is widely distributed in the elastic structure resulting in higher fracture resistance. Therefore, resin-containing restorations with an elastic modulus close to that of teeth may exhibit better fracture resistance when used in actual clinical practice compared to the results of extra-oral studies, where they are attached to rigid abutments.

When RNC was adhered to the TIs, the fracture resistance decreased significantly and exhibited a high probability of adhesion failure. It has been reported that RNC has lower adhesion to TIs compared to other CAD/CAM restorations [47,48]. Hydrofluoric acid etches the surface of the porcelain to create micro-porosities that facilitate micromechanical and chemical bonding between the ceramic and resin materials. Silane coupling agents promote adhesion and form a chemical bond with organic and inorganic surfaces, thereby increasing the wettability of the ceramic surfaces [49]. Consequently, the risk of fracture and detachment may be greater when RNC is used in an implant, though, it showed relatively high fracture resistance in teeth. In previous studies, it has been reported that the mean maximal bite force was 500 N with a range from 330 to 680 N. There were no statistically significant differences between the sexes regarding their maximal bite forces [50,51]. Therefore, LS and ZLS are expected to function well in all patients, but PICN and RNC are considered to be suitable for use in patients with relatively low occlusal force.

Fractographic studies using optical microscopy and scanning electron microscopy are used for the analysis of fracture propagation patterns [32,33,34,35]. Optical microscopes reveal irregularities in the surface well, but there are magnification limitations. FE-SEMs are useful for detecting small details of the fracture surfaces as they can be observed at higher magnifications, but they are not suitable for viewing the overall fracture shapes. Both methods play complementary roles [52].

The fracture resistance should also be coupled with the failure types [50]. In this study, the fracture resistances were significantly different in the four types of CAD/CAM crowns; however, no clear differences in fracture patterns were found between the materials. In almost all restorations made of a variety of materials, fracture lines originated from the contact point. Through observations of FE-SEM, a fracture mirror—an indicator of the most obvious origin of fracture—was found in several specimens. In most specimens, however, the fracture mirror was not discovered—in these situations, the origin point could be estimated by analyzing the propagation pattern of the fracture line. At the origin, a hackle—which means the shape of a bird’s neck hair—extended, revealing the propagation of the fracture [33]. In most circumstances, the fracture starting point was the area in which the ball indenter was in contact with the specimen. Furthermore, fractures started at the lower cement part under tension. The difference between the stressed area and the origin of fracture is that when a compressive force was applied to the occlusal surface, the tensile force was generated on the inner surface of the crown and fractures started at the cement interface.

The limitation of this study is that when standardizing TOs, different sizes of extracted teeth used standardize the shape of the mandibular first premolar. As the volume of removal increases, the modulus of elasticity may be lowered [52], which may have affected the differences in fracture resistances. In addition, when manufacturing abutments and crowns, errors can accumulate over multiple scans and milling processes. However, with the advancement of technology, the range of errors of scanners has been improved, and the scanner is now certified as a reliable device [53]. In the future, these errors will gradually decrease. Reproducing the clinical intraoral environment of approximately one year was attempted via thermal cycling treatment, which lacks the ability to reproduce the long-term oral environment and the occlusal load of the antagonist tooth. In order to analyze the results more accurately, additional studies that apply occlusal loads in the oral environment for prolonged periods of time are necessary.

## 6. Conclusions

Based on the findings of this in vitro study, the following conclusions were drawn.
For the fracture resistances of the restorations, there are statistically significant correlations between the types of chairside CAD/CAM materials and the types of abutments.Lithium disilicate and zirconia-reinforced lithium silicate restorations exhibited statistically significant high fracture resistances, indicating their suitability as restorative materials for natural teeth or implant abutments.The chairside CAD/CAM restorations showed different fracture failure modes based on the types of materials and abutments.The fracture of the chairside CAD/CAM restorations initiated at the groove where the ball indenter toughed and propagated toward the axial wall. There were no distinct differences based on the types of materials and abutments.

## Figures and Tables

**Figure 1 materials-14-03115-f001:**
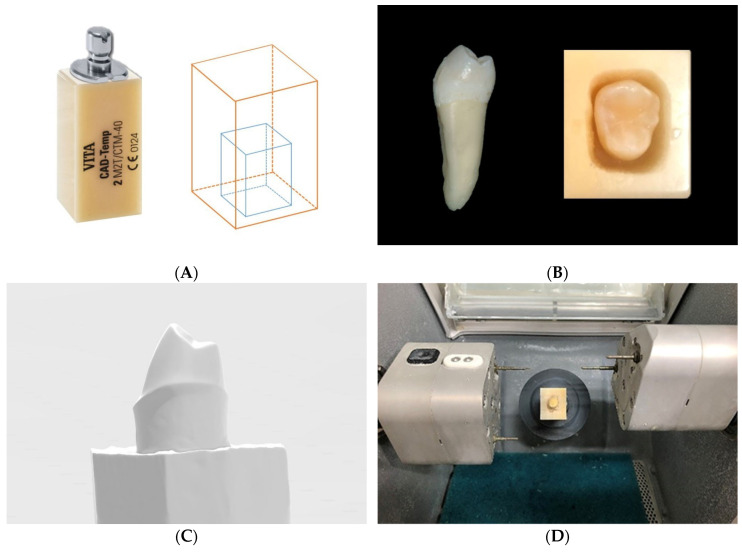
Standardization of tooth abutment: (**A**):VITA CAD-Temp multiColor block (left); schematic diagram for cutting block (right). (**B**): Mandibular premolar tooth (left); block with tooth placed inside (right). (**C**): The STL file of prepared teeth for standardization. (**D**): Tooth abutment fabricated using milling machine. (**E**): Standardized tooth abutments. (**F**): The STL file of the crown shape for standardization. (**G**): Specimen with crown cemented to tooth abutments. STL, standard tessellation language.

**Figure 2 materials-14-03115-f002:**
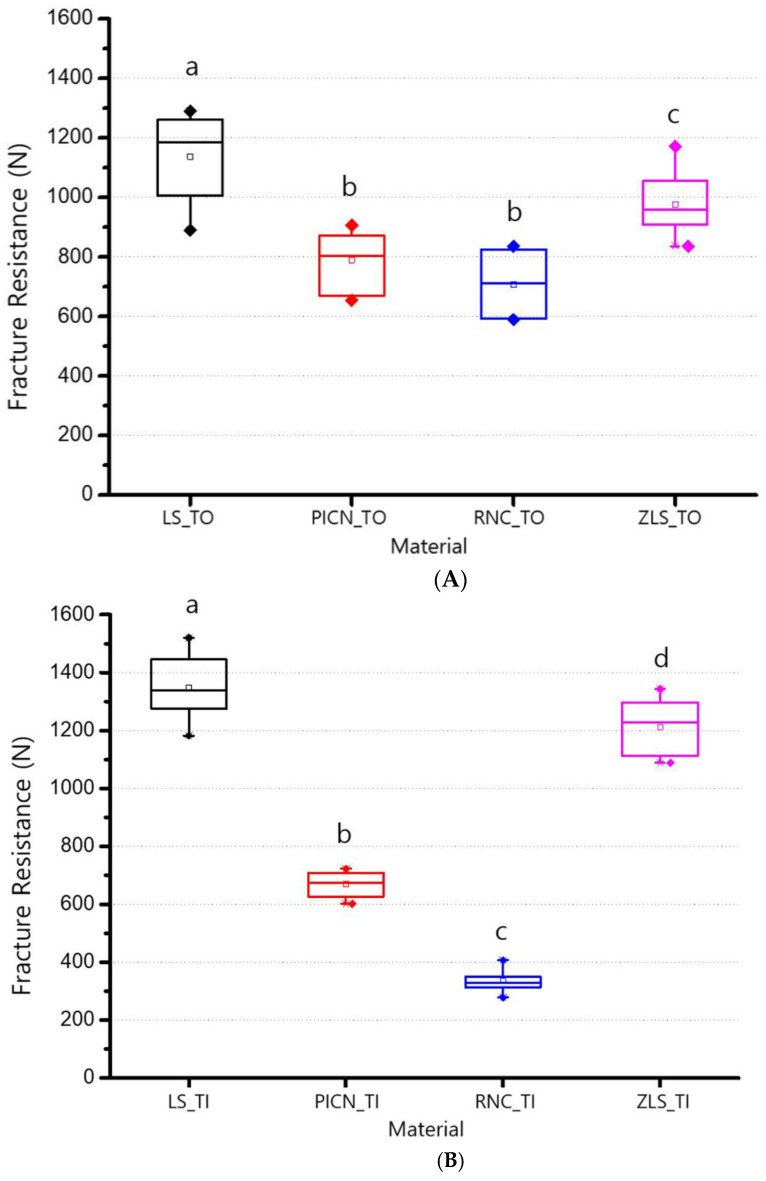
Means and standard deviations of fracture resistance. (**A**): Tooth abutment. (**B**): Titanium abutment. Same alphabet letters indicate no significant differences at *p* < 0.05. LS, lithium disilicate; PICN, polymer-infiltrated-ceramic-network; RNC, resin nano ceramic; ZLS, zirconia-reinforced lithium silicate; TO, tooth abutment; TI, titanium abutment.

**Figure 3 materials-14-03115-f003:**
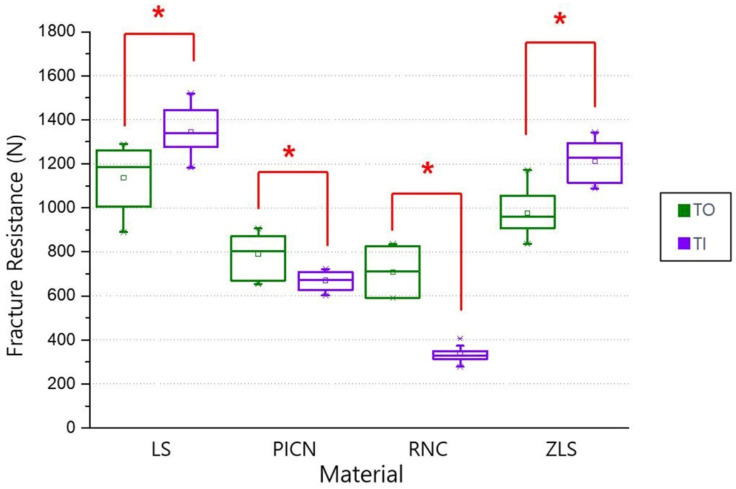
Means and standard deviations of fracture resistance. * Denotes significant differences at *p* < 0.05. LS, lithium disilicate; PICN, polymer-infiltrated-ceramic-network; RNC, resin nano ceramic; ZLS, zirconia-reinforced lithium silicate; TO, tooth abutment; TI, titanium abutment.

**Figure 4 materials-14-03115-f004:**
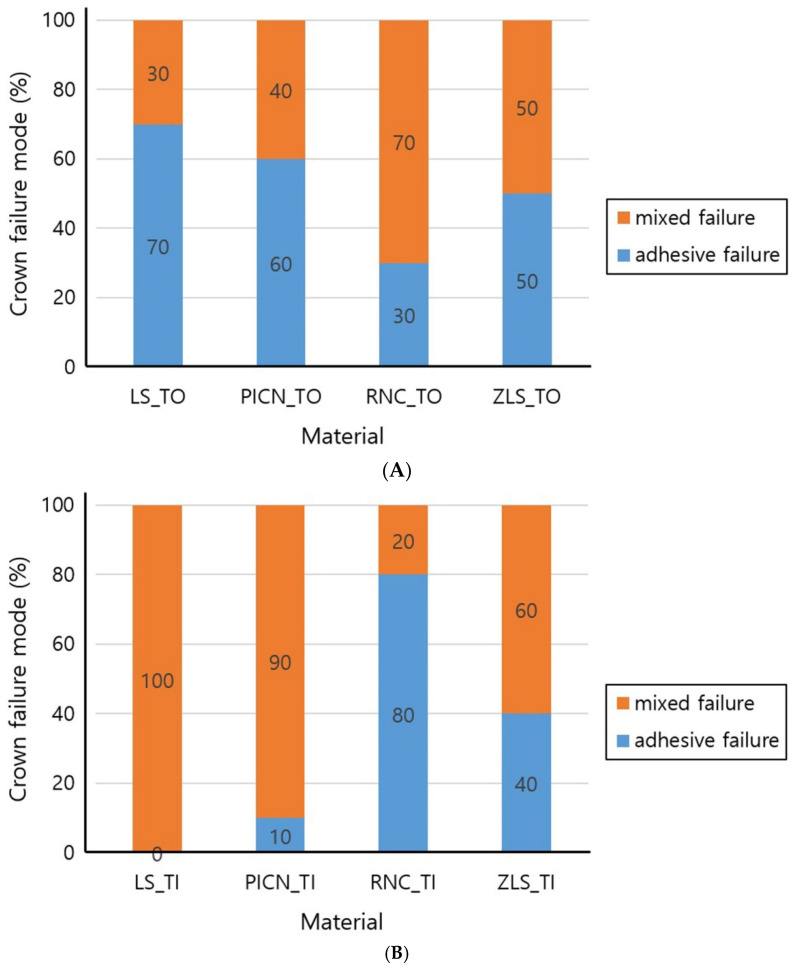
Crown failure mode (**A**): Tooth abutment. (**B**): Titanium abutment. LS, lithium disilicate; PICN, polymer-infiltrated-ceramic-network; RNC, resin nano ceramic; ZLS, zirconia-reinforced lithium silicate; TO, tooth abutment; TI, titanium abutment.

**Figure 5 materials-14-03115-f005:**
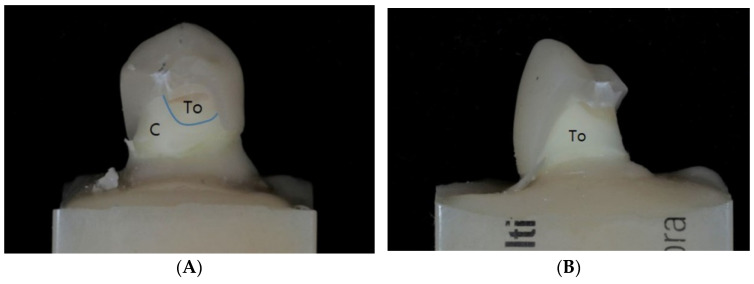
Fracture failure patterns in tooth and titanium abutments (**A**): PICN_TO specimen showing mixed failure pattern. (**B**): LS_TO specimen showing adhesive failure pattern. (**C**): LS_TI specimen showing mixed failure pattern. (**D**): RNC_TI specimen showing an adhesive failure pattern; the cement was attached to the abutment. (**E**): ZLS_TI specimen showing an adhesive failure pattern; the cement was attached to the crown fragment. PICN, polymer-infiltrated-ceramic-network; TO, tooth abutment; LS, lithium disilicate; TI, titanium abutment; RNC, resin nano ceramic; ZLS, zirconia-reinforced lithium silicate; C, cement; To, tooth; Ti, titanium.

**Figure 6 materials-14-03115-f006:**
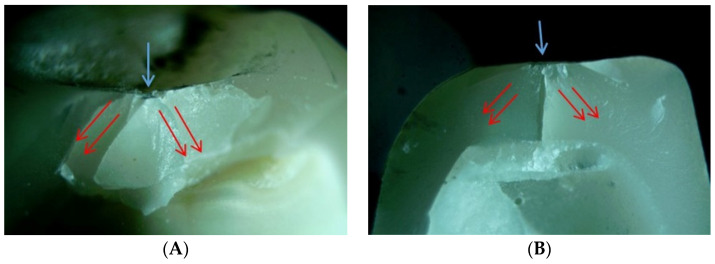
Stereomicroscopy images (25× magnification) of crowns corresponding to the load area (**A**): features of crown fracture of the representative crown from the PICN_TO group. (**B**): Features of crown fracture of the representative crown from the RNC_TO group. (**C**): Features of crown fracture of the representative crown from the ZLS_TO group. (**D**): Features of crown fracture of the representative crown from the LS_TO group. (**E**): Features of crown fracture of the representative crown from the PICN_TI group. (**F**) Features of crown fracture of the representative crown from the RNC_TI group. (**G**): Features of crown fracture of the representative crown from the ZLS_TI group. (**H**): Features of crown fracture of the representative crown from the LS_TI group. Blue arrows indicate the load areas; red arrows refer to the directions of the crack propagations. PICN, polymer-infiltrated-ceramic-network; TO, tooth abutment; RNC, resin nano ceramic; ZLS, zirconia-reinforced lithium silicate; LS, lithium disilicate; TI, titanium abutment.

**Figure 7 materials-14-03115-f007:**
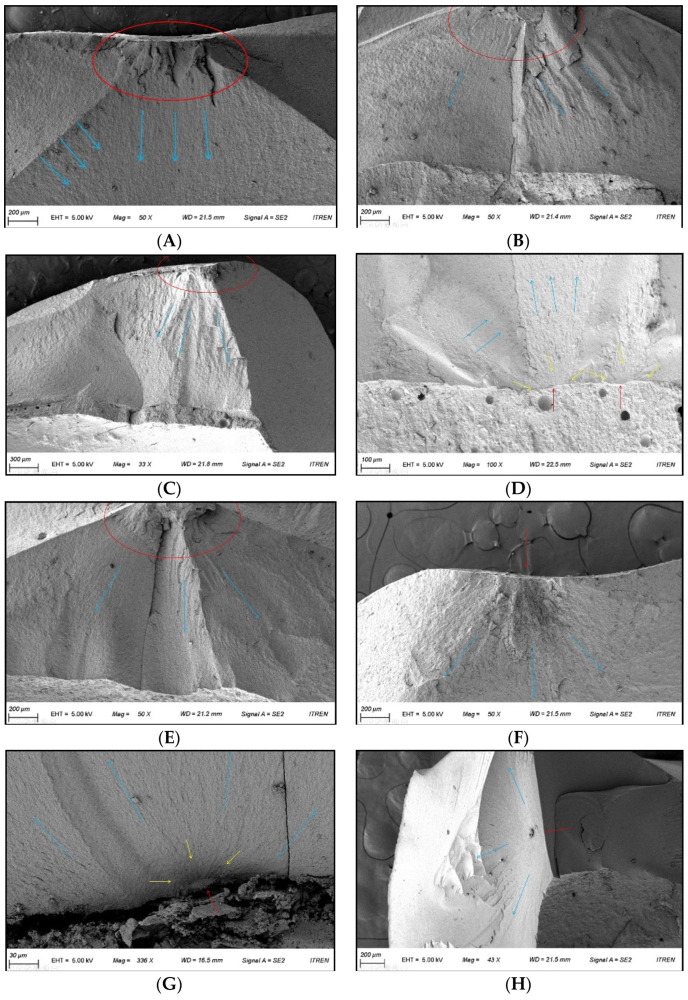
SEM image of crowns in tooth and titanium abutments corresponding to the area of crack origin. (**A**): Representative crown from group LS_TO (50× magnification). (**B**): Representative crown from group PICN_TO (50× magnification). (**C**): Representative crown from group RNC_TO (33× magnification). (**D**): Representative crown from group ZLS_TO (100× magnification). (**E**): Representative crown from group LS_TI (50× magnification). (**F**): Representative crown from group PICN_TI (50× magnification). (**G**): Representative crown from group RNC_TI (336× magnification). (**H**): Representative crown from group ZLS_TI (43× magnification). Red arrows and circles indicate the origin; yellow arrows refer to the ‘fracture mirror’; blue arrows refer to the ‘hackle’.

**Table 1 materials-14-03115-t001:** Composition and manufacturers information of the tested materials.

Product Name	Code	Manufacturer	Lot No.	Shade	Composition	Groups	
						Tooth	Titanium
IPS e.max CAD	LS	Ivoclar Vivadent	Y00999, Y26950	A2-HT	0.2–2 μm lithium disilicate glass-ceramic	LS_TO	LS_TI
Vita Enamic	PICN	Vita Zahnfabrik	78540, 78880	2M2-HT	Polymer-infiltrated feldspathic ceramic-network material (UDMA, TEGDMA) with 86 wt% ceramic	PICN_TO	PICN_TI
Cerasmart	RNC	GC dental product	1910101	A2-HT	Composite resin material (BisMEPP, UDMA, DMA) with 71 wt% silica and barium glass nanoparticles	RNC_TO	RNC_TI
Celtra Duo	ZLS	Dentsply Sirona	16006746, 16006750	A2-HT	10% dissolved zirconia in a silica-based glass matrix.	ZLS_TO	ZLS_TI

LS, lithium disilicate; PICN, polymer-infiltrated-ceramic-network; RNC, resin nano ceramic; ZLS, zirconia-reinforced lithium silicate; TO, tooth abutment; TI, titanium abutment; UDMA, urethane dimethacrylate; TEGDMA, triethylene glycol dimethacrylate; BisMEPP, 2,2-bis (4-methyacryloxypolyethoxyphenyl) propane; DMA, dodecyl dimethacrylate.

**Table 2 materials-14-03115-t002:** Mean ± SD values and statistical analysis of fracture resistance.

Group	Mean ± SD (N)		*p*
	TO	TI	
LS	1137.33 ± 139.30 ^a^	1346.60 ± 103.53 ^a^	0.001 *
PICN	789.73 ± 98.90 ^b^	670.24 ± 40.80 ^b^	0.002 *
RNC	707.39 ± 100.74 ^b^	334.39 ± 36.30 ^c^	<0.001 *
ZLS	976.47 ± 107.37 ^c^	1211.32 ± 93.70 ^d^	<0.001 *

SD, standard deviation; LS, lithium disilicate; PICN, polymer-infiltrated-ceramic-network; RNC, resin nano ceramic; ZLS, zirconia-reinforced lithium silicate; TO, tooth abutment; TI, titanium abutment. *p* values were calculated using the result of an independent samples *t*-test between the tooth and titanium groups. * Denotes a significant difference at *p* < 0.05. Values with different superscript letters in each vertical column indicate significant differences from each other (*p* < 0.05).

**Table 3 materials-14-03115-t003:** Results of two-way ANOVA.

Source	Type III Sum of Squares	df	Mean Square	F	*p*
Corrected Model	7,804,216.38 ^a^	7	1,114,888.055	121.937	<0.001 *
Intercept	64,323,143.29	1	64,323,143.29	7035.104	<0.001 *
Material	6,542,454.222	3	2,180,818.074	238.519	<0.001 *
Abutment	2923.299	1	2923.299	0.320	0.574
Material * abutment	1,258,838.862	3	419,612.954	45.894	<0.001 *
Error	658,308.150	72	9143.169		
Total	72,785,667.82	80			
Corrected Total	8,462,524.533	79			

^a^ R Squared = 0.922 (Adjusted R Squared = 0.915). * Denotes significant difference at *p* < 0.05.

## Data Availability

Data sharing not applicable.

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
