# Peer review of "Comparison of Mechanical Properties of Chairside CAD/CAM Restorations Fabricated Using a Standardization Method"

_materials, 2021, doi:10.3390/ma14113115_

Round 1

Reviewer 1 Report

This is a very interesting work on the fracture effects of different material systems obtained with the CAD CAM technique

The work is well structured and executed and innovative from a scientific point of view. Beautiful pictures and the explanation of the study.

Some criticisms are however present:

-The abstract section is too long. It must absolutely be restricted, not dwelling on the technical aspects of the study

-In the introduction section the sentence after the bibliographic reference 3,4 is not appropriate because a relationship between the CAD CAM technology and the in vitro studies is not indicative

- All the part from reference 14 to the end of the paragraph is useless and inappropriate in the introduction section because it must frame the problem and the materials, not contain reflections on the various methods of investigation

-The null hypotheses are in the plural, so fix the grammar

-In the initial part of the discussion a concise sentence on the results obtained should be inserted

- A paragraph is missing referring to the limits of the study that must be inserted

-In the discussion section, some considerations should also be made on the predictability of oral scanner systems, a fundamental element for the reliability of the measurements. In this regard, I recommend that you include the following scientific work in the reference section that could be of help to the reader:

Pagano S, Moretti M, Marsili R, Ricci A, Barraco G, Cianetti S. Evaluation of the Accuracy of Four Digital Methods by Linear and Volumetric Analysis of Dental Impressions. Materials (Basel). 2019 Jun 18; 12 (12): 1958. doi: 10.3390 / ma12121958. PMID: 31216639; PMCID: PMC6631156.

-Tables 1 and 2 should be merged into a single table

-A certified check of the English language is required by this reviewer

Author Response

<Comments and Suggestions for Authors
-1>

Point : This is a very interesting work on the fracture effects of different material systems obtained with the CAD CAM technique.

The work is well structured and executed and innovative from a scientific point of view. Beautiful pictures and the explanation of the study.

Response : Thank you for your kind comments.

Some criticisms are however present:

Point 1 : The abstract section is too long. It must absolutely be restricted, not dwelling on the technical aspects of the study

Response 1 : Thank you for this suggestion. We have now shortened the abstract section.

We have changed the first paragraph “With the rapid development of computer-aided design/computer-aided manufacturing (CAD-CAM) technology, new dental materials are being created. However, in vitro studies that reproduce the oral environment are insufficient and there are limitations in the analysis of results owing to the non-standardized form of extracted teeth. In this study, the fracture resistances of four types of chairside CAD-CAM restorations in tooth and titanium abutments fabricated using a standardized method were compared, the fracture and propagation patterns of which were analyzed.” to “The aim of this in vitro study was to investigate the fracture resistances and fractography of chairside computer-aided design/computer-aided manufacturing (CAD/CAM) restorations in tooth and titanium abutments fabricated using standardization method.”

And we have revised “and statistical analysis was performed. The differences in fracture resistance between the restorations were evaluated using a one-way analysis of variance (ANOVA), and the differences in fracture resistance by abutment for the same restorations were evaluated using an independent t-test. The correlation between titanium and dental abutments and CAD-CAM restorations was analyzed using a two-way ANOVA (α=.05).” to “Statistical differences were analyzed using appropriate ANOVA, Tukey HSD post hoc tests, and independent sample t-tests (α=.05).”

Point 2 : In the Introduction section the sentence after the bibliographic reference 3,4 is not appropriate because a relationship between the CAD CAM technology and the in vitro studies is not indicative

Response 2 : Thank you for the comment. We acknowledge that there is an error in the part that shows the correlation between CAD/CAM technology and in vitro experiments. It has been modified as follows; “This technology facilitates the design of specimens for in vitro studies that evaluate the function of dental materials. In addition, the technology is also used for standardizing and manufacturing in a specific form.”

Point 3 : All the part from reference 14 to the end of the paragraph is useless and inappropriate in the Introduction section because it must frame the problem and the materials, not contain reflections on the various methods of investigation

Response 3 : Thank you for the comment. We have revised all the part from reference 14 to the end of the paragraph.

“The physical properties of the materials used in the fabrication of these chairside CAD/CAM restorations have been studied using various methods. However, to understand the physical properties of the material itself, most studies have been conducted by processing blocks in the form of cuboids or cylinders [9,14–17]. Artificial materials can be produced uniformly in the same shape, but human teeth cannot be unified because they all look different. In order to collect systematic information, it is necessary to standardize teeth as a specimen. However, standardizing teeth into the same shape requires very complex processes, and studies that have attempted such procedures are insufficient. Therefore, we devised a method for standardization of teeth using CAD/CAM technology. Titanium abutments can also be milled into the same shape to create an opportunity to compare natural teeth and implants. [18-20].

In addition, studies on the fracture resistance of the upper structure due to the different elasticity modulus of the abutments are also scarce [21,22]. Also, we were interested in the analysis of fracture patterns and fractography of these chairside CAD/CAM materials. In general, adhesion is used in teeth and cementation is used in titanium abutments [23,24]. In this regard, we tried to analyze the pattern of fracture failure. Furthermore, since fractography can be observed after the material is fractured, there are few studies using in vivo restorations. As a preliminary step toward in vivo research, we tried to analyze the fracture pattern under standardized conditions in vitro.”

Point 4 : The null hypotheses are in the plural, so fix the grammar

Response 4 : Thank you for the comment. We have corrected the grammar as follows; “The null hypotheses proposed in this study are”

Point 5 : In the initial part of the discussion a concise sentence on the results obtained should be inserted

Response 5 : Thank you for this suggestion. We have inserted the following sentence in the initial part of the Discussion Section.

“In this study, TIs and TOs were standardized using CAD/CAM technology, and physical properties of four different chairside CAD/CAM materials were compared: LS, ZLS, PICN, and RNC. There were statistically significant differences in fracture resistance, with the highest being LS and the lowest being RNC. Each of the different CAD/CAM materials showed different fracture failure patterns, and there was no noticeable difference in fractographic analysis.”

Point 6 : A paragraph is missing referring to the limits of the study that must be inserted

Response 6 : Thank you for the comment. We have added and revised the limitations at the end of the Discussion Section as follows;

“The limitation of this study is that when standardizing TOs, extracted teeth of various sizes were used for standardization of the mandibular first premolar. As the volume of removal increases, the modulus of elasticity may be lowered [52], which is related to the differences in fracture resistance. Moreover, when manufacturing abutments and crowns, errors can accumulate over multiple scans and milling processes. However, with the advancement of technology, the range of errors of the scanner have been improved, and the scanner is certified as a reliable device [53,54]. In the future, these errors will gradually decrease. Furthermore, it was attempted to reproduce the clinical intraoral environment of approximately one year by using a thermal cycling treatment, which lacks the ability to reproduce the long-term oral environment and the occlusal load of the antagonist tooth. In order to analyze the results more accurately, additional studies that apply occlusal loads in the oral environment for prolonged periods of time are necessary.”

Point 7 :  In the discussion section, some considerations should also be made on the predictability of oral scanner systems, a fundamental element for the reliability of the measurements. In this regard, I recommend that you include the following scientific work in the reference section that could be of help to the reader:

Pagano S, Moretti M, Marsili R, Ricci A, Barraco G, Cianetti S. Evaluation of the Accuracy of Four Digital Methods by Linear and Volumetric Analysis of Dental Impressions. Materials (Basel). 2019 Jun 18; 12 (12): 1958. doi: 10.3390 / ma12121958. PMID: 31216639; PMCID: PMC6631156.

Response 7 : Thank you for your kind suggestion. We have added the reference in the last paragraph of Discussion Section as follow; “However, with the advancement of technology, the range of errors of the scanner have been improved, and the scanner is certified as a reliable device [53,54]. In the future, these errors will gradually decrease.”

References

  1. Pagano, S.; Moretti, M.; Marsili, R.; Ricci, A.; Barraco, G.; Cianetti, S. Evaluation of the Accuracy of Four Digital Methods by Linear and Volumetric Analysis of Dental Impressions. Materials (Basel). 2019, 12, 1958.
  2. Son, K.; Lee, K.B. Effect of Tooth Types on the Accuracy of Dental 3D Scanners: An In Vitro Study. Materials (Basel). 2020, 13, 1744.

Point 8 : Tables 1 and 2 should be merged into a single table

Response 8 : Thank you for the comment. We have merged Tables 1 and 2.

Table 1. Composition and manufacturers information of tested materials

Product

name

Code

Manufacturer

Lot no.

Shade

Composition

Groups

Tooth

Titanium

IPS e.max CAD

LS

Ivoclar

Vivadent

Y00999,

Y26950

A2-HT

0.2-2 μm lithium disilicate

glass-ceramic

LS_TO

LS_TI

Vita

Enamic

PICN

Vita

Zahnfabrik

78540,

78880

2M2-HT

Polymer-infiltrated feldspathic ceramic-network material

(UDMA, TEGDMA) with 86 wt% ceramic

PICN_TO

PICN_TI

Cerasmart

RNC

GC dental

product

1910101

A2-HT

Composite resin material

(BisMEPP, UDMA, DMA)

with 71 wt% silica and

barium glass nanoparticles

RNC_TO

RNC_TI

Celtra

Duo

ZLS

Dentsply

Sirona

16006746,

16006750

A2-HT

10% dissolved zirconia in a

silica-based glass matrix.

ZLS_TO

ZLS_TI

LS, lithium disilicate; PICN, polymer-infiltrated-ceramic-network; RNC, resin nano-ceramic; ZLS, zirconia-reinforced lithium silicate; TO, tooth abutment; TI, titanium abutment; UDMA, urethane dimethacrylate; TEGDMA, triethylene glycol dimethacrylate; BisMEPP, 2,2-bis (4-methyacryloxypolyethoxyphenyl) propane; DMA, dodecyl dimethacrylate.

Point 9 : A certified check of the English language is required by this reviewer

Response 9 : Thank you for the comment. Per your suggestion, the manuscript has now been carefully revised to improve its readability, in addition to having been checked by Elsevier Language Editing Service to improve the level of English.

Reviewer 2 Report

The manuscript seems to present an interesting issue, but it is poorly written. There are many aspect to be improved. The most important of them are the aim formulation and the description of study design.

Please add line numbers.

Introduction

Please explain in detail why this study is important. Please add background for the study, is it important in cementation of implant restorations?

After reading introduction and aim I did not know what is the purpose.

The third paragraph of the introduction should be summarised and moved to discussion.

The fourth paragraph should be moved to method section.

Aim

Please state an aim in more clear way, now it is misleading. I suggest writing it as points.

State H0 as points.

Methods

Explain all the abbreviation when used for the first time in text, i.e. what TO and TI stands for?

Sample preparation should be presented in a more clear way – you might prepare a graph.

Why natural teeth were needed?

What were materials used for abutments?

How crowns were cemented?

Results

Please move all the figures and tables under paragraph where they were cited.

Describe in text the most important findings, especially statistically significant differences.

Discussion

What is the clinical significance of the study?

Is it possible to extrapolate conditions used in the study on the clinical situation?

If yes, this should be added (describe in detail) in the introduction.

If not, should be widely discussed comparing with real clinical scenario.

Please discuss influence of different defections of occlusal loads.

Please discuss fracture failure mode analysis.

What are the other limitations of the study?

Conclusions

Conclusions should be corrected.

Conclusion 1 is not supported by the results

Conclusion 2 is the result

Conclusion 5 is not supported by the results

Author Response

<Comments and Suggestions for Authors
-2>

Point : The manuscript seems to present an interesting issue, but it is poorly written. There are many aspect to be improved. The most important of them are the aim formulation and the description of study design.

Please add line numbers.

Response : Thank you for your comments. Per your suggestion, the manuscript has now been carefully revised to improve its readability. And we have added line numbers.

Introduction

Point 1 : Please explain in detail why this study is important. Please add background for the study, is it important in cementation of implant restorations?

Response 1 : Thank you for the comment. To explain the points of this study, we have revised the Introduction Section as follows;

“The mechanical properties of the materials used in the fabrication of these chairside CAD/CAM restorations have been studied using various methods. However, to understand the mechanical properties of the material itself, most studies have been conducted by processing blocks in the form of cuboids or cylinders [9,14–17]. Artificial materials can be produced uniformly in the same shape, but human teeth cannot be unified because they all look different. In order to collect systematic information, it is necessary to standardize teeth as a specimen. However, standardizing teeth into the same shape requires very complex processes, and studies that have attempted such procedures are insufficient. Therefore, we devised a method for standardization of teeth using CAD/CAM technology. Titanium abutments can also be milled into the same shape to create an opportunity to compare natural teeth and implants. [18-20].

In addition, studies on the fracture resistance of the upper structure due to the different elasticity modulus of the abutments are also scarce [21,22]. Also, we were interested in the analysis of fracture patterns and fractography of these chairside CAD/CAM materials. In general, adhesion is used in teeth and cementation is used in titanium abutments [23,24]. In this regard, we tried to analyze the pattern of fracture failure. Furthermore, since fractography can be observed after the material is fractured, there are few studies using in vivo restorations. As a preliminary step toward in vivo research, we tried to analyze the fracture pattern under standardized conditions in vitro.”

Point 2 : After reading introduction and aim I did not know what is the purpose.

Response 2: Thank you for the comment. The following content has been added to the Introduction Section.

We focused on standardizing these differently shaped natural teeth into exactly the same shape. For that, we designed using CAD technology and milled natural teeth in the same shape. Using the abutment made in this way, we tested the properties of the crowns made of various types of materials, and tried to achieve an effect similar to that of a real oral experiment. In addition, the titanium abutments were also milled into the same shape, making it an opportunity to compare natural teeth and implants In addition, due to the characteristics of fractography, since it can be observed after the material is fractured, there have not been many studies on restorations that are actually used in clinical practice. Therefore, as a preliminary step in in vivo research, we tried to analyze the fracture pattern under standardized conditions.”

Point 3 : The third paragraph of the introduction should be summarised and moved to discussion.

Response 3 : Thank you for the comment. We have removed that paragraph to the following, and the details are covered in the Discussion Section. We have revised the Introduction Section as follows;

“The mechanical properties of the materials used in the fabrication of these chairside CAD/CAM restorations have been studied using various methods. However, to understand the mechanical properties of the material itself, most studies have been conducted by processing blocks in the form of cuboids or cylinders [9,14–17]. Artificial materials can be produced uniformly in the same shape, but human teeth cannot be unified because they all look different. In order to collect systematic information, it is necessary to standardize teeth as a specimen. However, standardizing teeth into the same shape requires very complex processes, and studies that have attempted such procedures are insufficient. Therefore, we devised a method for standardization of teeth using CAD/CAM technology. Titanium abutments can also be milled into the same shape to create an opportunity to compare natural teeth and implants. [18-20].

In addition, studies on the fracture resistance of the upper structure due to the different elasticity modulus of the abutments are also scarce [21,22]. Also, we were interested in the analysis of fracture patterns and fractography of these chairside CAD/CAM materials. In general, adhesion is used in teeth and cementation is used in titanium abutments [23,24]. In this regard, we tried to analyze the pattern of fracture failure. Furthermore, since fractography can be observed after the material is fractured, there are few studies using in vivo restorations. As a preliminary step toward in vivo research, we tried to analyze the fracture pattern under standardized conditions in vitro.”

Point 4 : The fourth paragraph should be moved to method section.

Response 4 : Thank you for the comment. We have moved “Next, the mechanical properties were evaluated through a fracture test, the failure patterns were analyzed, the fracture sections were observed, and the fracture patterns and their causes were analyzed” to the beginning of subheadings 2.2, 2.3, and 2.4 of the Materials and Methods Section, respectively.

Aim

Point 5 : Please state an aim in more clear way, now it is misleading. I suggest writing it as points.

Response 5 : Thank you for the comment. We have revised and added the aim of this study in the Introduction section to clarify it;

“The aim of this study is to standardize the teeth abutments (TOs) and titanium abutments (TIs) in the same shape for more precise and uniform data collection and to compare the mechanical properties of four types of chairside CAD/CAM restorations based on the types of abutments.”

Point 6 : State H0 as points.

Response 6 : Thank you for the comment. We have revised the null hypothesis “First, there is no difference in the mechanical properties between natural TOs and various chairside CAD/CAM restorations cemented to TIs. Second, there is no difference in the mechanical properties based on the type of abutment in each of the same chairside CAD/CAM restorations. Third, there is no interaction between various chairside CAD/CAM restorations and types of abutments.” to “First, there is no difference in the mechanical properties between various chairside CAD/CAM restorations cemented to TOs and TIs. Second, in each of the same CAD/CAM material, there is no difference in the mechanical properties based on the type of abutment. Third, there is no interaction between various chairside CAD/CAM restorations and types of abutments.”

Material & Methods

Point 7 : Explain all the abbreviation when used for the first time in text, i.e. what TO and TI stands for?

Response 7 : Thank you for the comment. We have stated all the abbreviation when used for the first time in text (i.e.) in the Introduction section as follows;

“The aim of this study is to standardize the teeth abutment (TOs) and titanium abutments (TIs) in the same shape for more precise and uniform data collection and to compare the mechanical properties of four types of chairside CAD/CAM restorations based on the types of abutments.”

Point 8 : Sample preparation should be presented in a more clear way – you might prepare a graph.

Response 8 : Thank you for the comment. We have now shown figures of the experiment process as follows;

Figure 1. Standardization of tooth abutment. A: VITA CAD-Temp multiColor block (left), Schematic

diagram for cutting block (right), B: Mandibular premolar tooth (left), Block with tooth placed inside (right), C: the STL file of prepared teeth for standardization. D: Tooth abutment fabricated using

milling machine. E: Standardized tooth abutments. F: The STL file of the crown shape for

B

A

standardization. G: Specimen with crown cemented to tooth abutments.

E

F

D

C

G

STL, standard tessellation language

Point 9 :  Why natural teeth were needed?

Response 9 : Thank you for the comment. We focused on standardizing these differently shaped natural teeth into exactly the same shape. For that, we designed using CAD technology and milled natural teeth in the same shape. Also, we was interested in the analysis of fracture patterns and fractography of these chairside CAD/CAM materials. However, due to the characteristics of fractography, since it can be observed after the material is fractured, there have not been many studies on restorations that are actually used in clinical practice. Therefore, we tried to analyze the fracture pattern under standardized conditions.

Point 10 :  What were materials used for abutments?

Response 10 : Thank you for the comment. 40 extracted mandibular premolars or molars and 40 titanium abutments were used. We have now added the contents.

“To compare the mechanical properties of TOs and TIs, TIs were fabricated by milling 40 titanium blocks in the same form. They were fabricated by CNC milling of premilled cylinder grade 5 titanium alloy (Ti-6Al-4V) with ARUM 5X-200 (Arum Europe GmbH, Frankfurt, Germany) : number of axis 5; accuracy 5 µm; spindle power DC 3.0 KW; spindle speed 2,000 – 60,000 rpm; ATC (automatic tool changer) number of tools 15 [26].”

Reference

  1. Yi, Y.; Heo, S.J.; Koak, J.Y.; Kim, S.K. Comparison of CAD/CAM abutment and prefabricated abutment in Morse taper internal type implant after cyclic loading: Axial displacement, removal torque, and tensile removal force. J. Adv. Prosthodont. 2019, 11, 305-312.

Point 11 : How crowns were cemented?

Response 11 : Thank you for the comment. They were cemented according to the manufacturer's instructions using Panavia F 2.0 (Kuraray, Tokyo, Japan) on the TOs and Tis as follows. We have now added the following to the text.

“Surface cleansing of crowns and abutments was performed and the surface of the crowns and TIs were sandblasted with 50 μm alumina powder at an air pressure of 0.1-0.4 MPa (14-58 PSI). After sandblasting, they were cleaned using an ultrasonic device for 2 minutes and then dried with a stream of air. Surface pretreatment was performed according to the manufacturer's instructions, and the restorations were cemented with a pressure of 50 N using a Dynamometer (NK-200, HANDPI, Zhejiang, China)[27]. Excess paste was removed and finished.”

Reference

  1. Habib, S.R.; Ansari, A.S.; Bajunaid, S.O.; Alshahrani, A.; Javed, M.Q. Evaluation of Film Thickness of Crown Disclosing Agents and Their Comparison with Cement Film Thickness after Final Cementation. Eur. J. Dent. 2020, 14, 224-232.

Results

Point 12 : Please move all the figures and tables under paragraph where they were cited.

Response 12 : Thank you for the comment. We have now moved the figures and tables under paragraph where they were cited.

Point 13 : Describe in text the most important findings, especially statistically significant differences.

Response 13 : Thank you for the comment. We have revised the subheadings 4.1. part in the Result Section particularly. We have indicated statistically significant differences.

We have revised “According to the one-way ANOVA analysis, the fracture resistance of PICN and RNC in the TO group did not show any significant difference (P>.05), but the other restorations did (P<.05) (Figure 2A). In the TI group, there were significant differences among all restorations (Figure 2B). According to the independent sample t-test analysis, the fracture resistance values based on TO and TI were significantly different in all CAD/CAM restorations. The fracture resistances of PICN and RNC were significantly lower in TI than in TO, and the fracture resistances of ZLS and LS were significantly higher in TI than in TO (Figure 3). Two-way ANOVA analysis revealed significant interactions between the four types of chairside CAD/CAM restorations, teeth, and TIs (Table 4).” to “According to the one-way ANOVA analysis, in the TO group, the fracture resistance of PICN and RNC did not show any significant difference (P>.05), but in the other restorations except between PICN and RNC, there were significant differences (P<.05) (Figure 2A). In the TI group, there were significant differences among all restorations (P<.05) (Figure 2B). According to the independent sample t-test analysis, the fracture resistance values based on TO and TI were significantly different in all CAD/CAM restorations (P<.05). In PICN and RNC group, the fracture resistances in TO were significantly higher than in TI. On the other hand, in ZLS and LS group, the fracture resistances of TI were significantly higher than in TO (Figure 3). However, the fracture resistance of LS_TO and ZLS_TO, which showed lower values compared to the values of TI, showed higher values than PICN_TO and RNC_TO, which showed higher values compared to the values of TI. Thus, the overall fracture resistance appeared in the descending order of LS_TI, ZLS_TI, LS_TO, ZLS_TO, PICN_TO, RNC_TO, PICN_TI, and RNC_TI. Two-way ANOVA analysis revealed significant interactions between the four types of chairside CAD/CAM restorations, teeth, and TIs (Table 3).”

Discussion

Point 14 : What is the clinical significance of the study?

Response 14 : Thank you for the comment. The clinical significance of the study is as follows;

LS and ZLS exhibited high fracture resistance, and excellent results can be expected as restorations for natural tooth or implant abutments. As PICN and RNC exhibited little difference to the modulus of elasticity of teeth, it is expected that they may show better results when used as restorations for natural abutments. They are particularly useful for patients with severe wear, taking advantage of their own low tendency to wear down antagonist teeth.

Point 15 : Is it possible to extrapolate conditions used in the study on the clinical situation?

If yes, this should be added (describe in detail) in the Introduction.

If not, should be widely discussed comparing with real clinical scenario.

Response 15 : Thank you for the comment. It is possible to extrapolate conditions used in the study on the clinical situation. Through fracture test, we could know the relationship between the abutment and the upper materials, and fractography can be used for in vivo experiments.

Per your suggestion, we have revised the Introduction Section as follows;

 “The mechanical properties of the materials used in the fabrication of these chairside CAD/CAM restorations have been studied using various methods. However, to understand the mechanical properties of the material itself, most studies have been conducted by processing blocks in the form of cuboids or cylinders [9,14–17]. Artificial materials can be produced uniformly in the same shape, but human teeth cannot be unified because they all look different. In order to collect systematic information, it is necessary to standardize teeth as a specimen. However, standardizing teeth into the same shape requires very complex processes, and studies that have attempted such procedures are insufficient. Therefore, we devised a method for standardization of teeth using CAD/CAM technology. Titanium abutments can also be milled into the same shape to create an opportunity to compare natural teeth and implants. [18-20].

In addition, studies on the fracture resistance of the upper structure due to the different elasticity modulus of the abutments are also scarce [21,22]. Also, we were interested in the analysis of fracture patterns and fractography of these chairside CAD/CAM materials. In general, adhesion is used in teeth and cementation is used in titanium abutments [23,24]. In this regard, we tried to analyze the pattern of fracture failure. Furthermore, since fractography can be observed after the material is fractured, there are few studies using in vivo restorations. As a preliminary step toward in vivo research, we tried to analyze the fracture pattern under standardized conditions in vitro.”

Point 16 :  Please discuss influence of different defections of occlusal loads.

Response 16 : Thank you for the comment. We have added the followings at the sixth paragraph in the Discussion Section

“In previous studies, it has been reported that the mean maximal bite force was 500 N with a range from 330 to 680 N [50,51]. There were no statistically significant differences between the sexes as regards their maximal bite forces. Therefore, LS and ZLS are expected to function well in all patients, but PICN and RNC are considered to be suitable for use in patients with relatively low occlusal force.”

References

  1. Floystrand, F.; Kleven, E.; Oilo, G. A novel miniature bite force recorder and its clinical application. Acta. Odontol. Scand. 1982, 40, 209-214.
  2. Attia, A.; Kern, M. Fracture strength of all-ceramic crowns luted using two bonding methods. J. Prosthet. Dent. 2004, 91, 247-252.

Point 17 : Please discuss fracture failure mode analysis.

Response 17 : Thank you for the comment. In the fracture pattern, both adhesive fracture and mixed fracture were occurred, but there were exceptionally many adhesive fractures in the RNC attached to the TO. We have explained in the discussion as follows;

“When RNC was adhered to the TIs, the fracture resistance decreased significantly and exhibited a high probability of adhesion failure. It has been reported that RNC has lower adhesion to abutments and cement than other CAD/CAM restorations [47,48]. The hydrofluoric acid etches the surface of the porcelain to create micro-porosities to facilitate micromechanical and chemical bonding between the ceramic materials and the resin materials. Silane coupling agents promote adhesion and form a chemical bond with organic and inorganic surfaces, thereby increasing the wettability of the ceramic surfaces [49]. Consequently, the risk of fracture and de-attachment is increased when RNC is used in an implant.”

References

  1. Krejci, I.; Daher, R.;. Stress distribution difference between Lava Ultimate full crowns and IPS e. max CAD full crowns on a natural tooth and on tooth-shaped implant abutments. Odontology. 2017, 105, 254–256.
  2. Schepke, U.; Lohbauer, U.; Meijer, H.J.; Cune, M.S. Adhesive failure of Lava Ultimate and Lithium Disilicate crowns bonded to zirconia abutments: A prospective within-patient comparison. Int. J. Prosthodont. 2018, 31, 208-210.
  3. Kumchai, H.; Juntavee, P.; Sun, A.F.; Nathanson, D. Comparing the Repair of Veneered Zirconia Crowns with Ceramic or Composite Resin: An in Vitro Study. Dent. J (Basel). 2020, 8, 37.

Point 18 : What are the other limitations of the study?

Response 18: Thank you for the comment. We have added and revised the other limitations at the end of the Discussion as follows;

“The limitation of this study is that when standardizing TOs, extracted teeth of various sizes were used for standardization of the mandibular first premolar. As the volume of removal increases, the modulus of elasticity may be lowered [52], which is related to the differences in fracture resistance. Moreover, when manufacturing abutments and crowns, errors can accumulate over multiple scans and milling processes. However, with the advancement of technology, the range of errors of the scanner have been improved, and the scanner is certified as a reliable device [53,54]. In the future, these errors will gradually decrease. Furthermore, it was attempted to reproduce the clinical intraoral environment of approximately one year by using a thermal cycling treatment, which lacks the ability to reproduce the long-term oral environment and the occlusal load of the antagonist tooth. In order to analyze the results more accurately, additional studies that apply occlusal loads in the oral environment for prolonged periods of time are necessary.”

Conclusions

Point 19 : Conclusions should be corrected.

Conclusion 1 is not supported by the results

Conclusion 2 is the result

Conclusion 5 is not supported by the results

Response 19 : Thank you for the comment. We have revised the Conclusion as follows.

“1. The fracture resistance of chairside CAD/CAM restorations was affected by the types of abutments.

  1. Lithium disilicates and zirconia-reinforced lithium silicates exhibited high fracture resistance, and excellent results are expected as restoration materials for natural teeth or implant abutments.
  2. Different fracture failure modes were revealed depending on the restoration materials and the types of abutments.
  3. There were no distinct differences in the fracture patterns depending on the restorations and abutment materials.”

Round 2

Reviewer 2 Report

The manuscript was not improved in a way allowing to be published in the present form. Therefore, I suggest rejection.

Please find below my major recommendations:

Newly added sections demand English editing.

Please add line numbers. They are not still in the manuscript.

The aim should be rewritten.

The null hypotheses are not clearly stated.

Materials and methods

Please describe this section presenting steps in the sequence that they were performed.

The cementation protocol is still not clear. Did you use the primer? Was enamel etched?

Please change Theory/calculation to Statistical analysis

Results

Results are not clearly presented. Only statistically significant differences should be described.

The subtitle of a figure should be under a figure.

Present figures as whiskers and box plots.

Please remove table 4 as it presents the same data that are in figures 4 and 5.

Please remove fragments describing methods.

Discussion

This section demands rewriting. There are many statements that are not clear of false.

The study limitations are poorly written.

Conclusions

Conclusions are still poorly written.

Author Response

<Comments and Suggestions for Authors>

Point. The manuscript was not improved in a way allowing to be published in the present form.

Therefore, I suggest rejection.

Response : Thank you for your comments. Per your suggestion, the manuscript has now been carefully revised to improve its readability, in addition to having been checked by MDPI English Editing Service to improve the level of English.

Please find below my major recommendations:

Point 1. Newly added sections demand English editing.

Response 1: Thank you for your comment. Per your suggestion, the manuscript has now been carefully revised to improve its readability, in addition to having been checked by MDPI English Editing Service to improve the level of English.

Point 2. Please add line numbers. They are not still in the manuscript.

Response 2: Thank you for your comments. Per your suggestion, we have now added line numbers in the manuscript.

Point 3. The aim should be rewritten.

Response 3: Thank you for your comment. We have revised the aim “The aim of this study is to standardize the teeth abutments (TOs) and titanium abutments (TIs) in the same shape for more precise and uniform data collection and to compare the mechanical properties of four types of chairside CAD/CAM restorations based on the types of abutments.” to “The aim of this in vitro study was to investigate the fracture resistances, fracture failure pattern and fractography of four types of chairside CAD/CAM restoration materials – LS, ZLS, PICN and RNC – in teeth abutments (TOs) and titanium abutments (TIs) fabricated using a standardization method.”

Point 4. The null hypotheses are not clearly stated.

Response 4: Thank you for your comment. We have now revised the null hypotheses as follows;

“The null hypotheses proposed in this study are as follows: First, there is no difference in the mechanical properties – fracture resistance, fracture failure pattern and fractography – between four types of chairside CAD/CAM restorations cemented to TOs and TIs. Second, in each sample of the same CAD/CAM material, there is no difference in the mechanical properties between TO and TI. Third, there is no correlation in the mechanical properties between types of CAD/CAM materials and the types of abutments.”

Materials and methods

Point 5. Please describe this section presenting steps in the sequence that they were performed.

Response 5: Thank you for your comment. Per your suggestion, we have now described the Materials and Methods Section in the sequence that we were performed as follows;

“First of all, we fabricated teeth abutments (TOs) and titanium abutments (TIs) using extracted teeth and titanium column by a standardization method. And crowns were manufactured with four types of CAD/CAM materials and cemented to TOs and TIs. The cementation process is described in more detail. After thermocycling of the specimens, the mechanical properties were evaluated. The fracture resistance, the fracture failure pattern, and the fractography were analyzed.”

Point 6. The cementation protocol is still not clear. Did you use the primer? Was enamel etched?

Response 6: Thank you for your comments. Per your suggestion, we have now described the cementation process in more detail to clarify it as follows;

“A total of 80 crowns were classified by material and the cementation process was carried out. Surface cleansing of crowns and abutments was performed. Then, the surface of the crowns and TIs were sandblasted with 50 μm alumina powder at an air pressure of 0.1–0.4 MPa (14–58 PSI), and they were cleaned using an ultrasonic device for 2 minutes, then dried with a stream of air. Surface pretreatment of crowns was performed according to the manufacturer's instructions using Panavia F 2.0 (Kuraray, Tokyo, Japan). Clearfil Ceramic Primer (Kuraray, Tokyo, Japan) were applied to the internal surface of the restorations. Panavia F 2.0 ED Primer Ⅱ (Kuraray, Tokyo, Japan) were applied to the surfaces of TOs. And then mixed Panavia F 2.0 Paste was applied and the restorations were cemented with a pressure of 50 N using a Dynamometer (NK-200, HANDPI, Zhejiang, China) [27]. Excess paste was removed and finished (Figure 1G). The specimens were dried at room temperature for 24 hours. The specimens were then classified into eight groups (N=10 for each group).”

Point 7. Please change Theory/calculation to Statistical analysis

Response 7: Thank you for your suggestion. Per your suggestion, we have now changed “Theory/calculation” to “Statistical analysis”.

Results

Point 8. Results are not clearly presented. Only statistically significant differences should be described.

Response 8: Thank you for your comments. We have now revised the Results Section as follows. Per your suggestion, we deleted the contents that were not statistically significant and tried to clearly describe the results.

“According to the one-way ANOVA analysis, in the TO group, LS (1137.33 ± 139.30 N) had the highest fracture resistance value, and the second was ZLS (976.47 ± 107.37 N) (p < 0.05). There was no statistically significant difference between PICN (789.73 ± 98.90 N) and RNC (707.39 ± 100.74 N) (p > 0.05; Figure 2A). In the TI group, LS (1346.60 ± 103.53 N) showed the highest value for fracture resistance, followed by ZLS (1211.32 ± 93.70 N), PICN (670.24 ± 40.80 N), with RNC (334.39 ± 36.30 N) showing the lowest value (p < 0.05; Figure 2B). According to the independent samples t-test, the fracture resistances based on TO and TI groups were significantly different in all the CAD/CAM restoration materials (p < 0.05). In the LS and ZLS groups, the fracture resistance values of TIs were significantly higher than the values of TOs (p < 0.05). Conversely, in the PICN and RNC groups, the fracture resistance values of TOs were significantly higher than the values of TIs (p < 0.05; Figure 3). Two-way ANOVA analysis revealed significant correlations between the four types of chairside CAD/CAM restorations and types of abutments. (p < 0.05; Table 3).”

Point 9. The subtitle of a figure should be under a figure.

Response 9: Thank you for your suggestion. Following your suggestion, we have now moved the subtitles below the figures in all figures.

Point 10. Present figures as whiskers and box plots.

Response 10: Thank you for your suggestion. Per your suggestion, we have now changed Figures 2 and 3 as whiskers and box plots.

Point 11. Please remove table 4 as it presents the same data that are in figures 4 and 5.

Response 11: Thank you for your suggestion. Per your suggestion, we have now removed Table 4.

Point 12. Please remove fragments describing methods.

Response 12: Thank you for your suggestion. Per your suggestion, we have now removed following fragments describing methods. “2.1. Preparation of specimens / 2.2. Fracture resistance test / 2.3. Fracture failure mode analysis / 2.4. Fractography analysis”

Discussion

Point 13. This section demands rewriting. There are many statements that are not clear of false.

Response 13: Thank you for these comments. Following your suggestion, we have carefully revised the Discussion Section and modified it overall. The manuscript has now been carefully revised to clarify the intent and to improve its readability.

Point 14. The study limitations are poorly written.

Response 14: Thank you for your comment. We have revised the study limitations to clarify the meaning and improve readability in the Discussion Section. If there is any content that needs further correction, let us know and we will correct it, any advice would be appreciated. We have revised the study limitations as follows;

“The limitation of this study is that when standardizing TOs, different sizes of extracted teeth were used standardize the shape of the mandibular first premolar. As the volume of removal increases, the modulus of elasticity may be lowered [52], which may have affected to the differences in fracture resistances. In addition, when manufacturing abutments and crowns, errors can accumulate over multiple scans and milling processes. However, with the advancement of technology, the range of errors of scanners has been improved, and the scanner is now certified as a reliable device [53,54]. In the future, these errors will gradually decrease. Reproducing the clinical intraoral environment of approximately one year was attempted via thermal cycling treatment, which lacks the ability to reproduce the long-term oral environment and the occlusal load of the antagonist tooth. In order to analyze the results more accurately, additional studies that apply occlusal loads in the oral environment for prolonged periods of time are necessary.”

Conclusions

Point 15. Conclusions are still poorly written.

Response 15: Thank you for your comment. We have revised the conclusions to clarify the meaning and to improve its readability. If there is any content that needs further correction, let us know and we will correct it, any advice would be appreciated. We have revised the Conclusion Section as follows;

“1. For the fracture resistances of the restorations, there are statistically significant correlations between the types of chairside CAD/CAM materials and the types of abutments.

  1. Lithium disilicate and zirconia-reinforced lithium silicate restorations exhibited statistically significant high fracture resistances, and indicating their suitability as restorative materials for natural teeth or implant abutments.
  2. The chairside CAD/CAM restorations showed different fracture failure modes based on the types of materials and abutments.
  3. The fracture of the chairside CAD/CAM restorations initiated at the groove where the ball indenter toughed and propagated toward the axial wall. There were no distinct differences based on the types of materials and abutments. There were no distinct differences based on the types of materials and abutments.”
